# Global warming-induced upper-ocean freshening and the intensification of super typhoons

Karthik Balaguru[1], Gregory R. Foltz[2], L. Ruby Leung[3] & Kerry A. Emanuel[4]

Super typhoons (STYs), intense tropical cyclones of the western North Pacific, rank among the most destructive natural hazards globally. The violent winds of these storms induce deep mixing of the upper ocean, resulting in strong sea surface cooling and making STYs highly sensitive to ocean density stratification. Although a few studies examined the potential impacts of changes in ocean thermal structure on future tropical cyclones, they did not take into account changes in near-surface salinity. Here, using a combination of observations and coupled climate model simulations, we show that freshening of the upper ocean, caused by greater rainfall in places where typhoons form, tends to intensify STYs by reducing their ability to cool the upper ocean. We further demonstrate that the strengthening effect of this freshening over the period 1961–2008 is ~53% stronger than the suppressive effect of temperature, whereas under twenty-first century projections, the positive effect of salinity is about half of the negative effect of ocean temperature changes.

[1] Marine Sciences Laboratory, Pacific Northwest National Laboratory, 1100 Dexter Avenue North, Seattle, Washington 98109, USA. [2] Physical Oceanography Division, Atlantic Oceanographic and Meteorological Laboratory, NOAA, Miami, Florida 33149, USA. [3] Atmospheric Sciences and Global Change, Pacific Northwest National Laboratory, Richland, Washington 99352, USA. [4] Lorenz Center, Massachusetts Institute of Technology, Cambridge, Massachusetts 02139, USA. Correspondence and requests for materials should be addressed to K.B. (email: Karthik.Balaguru@pnnl.gov).

The northwestern tropical Pacific Ocean is home to nearly a third of all tropical cyclones, making it the most active basin globally[1]. Tropical cyclones in the northwestern Pacific, commonly referred to as typhoons, are also some of the largest[2] and most destructive storms in the world, with impacts on many East Asian countries and Oceania islands. Super typhoons (STYs), equivalent in strength to an intense Category 4 or Category 5 hurricane, are particularly devastating. For instance, STY Haiyan made landfall in the Philippines in November 2013 as one of the most intense tropical cyclones in recorded history. The associated storm surge inflicted catastrophic damages, including nearly 6,000 fatalities and economic damages exceeding $800 million[3]. With many coastal areas in this region experiencing some of the highest rates of sea-level rise worldwide[4], better knowledge of future changes in typhoon intensities is pertinent.

As typhoons intensify by extracting energy from the warm ocean surface, the underlying sea surface temperatures (SSTs) are critical for their development[5]. The strong winds of typhoons induce vigorous mixing of the upper ocean, a process that tends to reduce SST through entrainment of colder water from below the surface mixed layer[6]. The resultant cooling of SST subsequently acts as a negative feedback on the typhoon's intensity through its impact on air–sea enthalpy fluxes[7–13]. Thus, the state of the upper ocean, including near-surface stratification, plays an important role in the intensification of typhoons. Near-surface ocean stratification is particularly important for the development of intense tropical cyclones, which are characterized by deep mixing of the upper ocean and strong sea surface cooling[14,15]. Hence, changes in upper-ocean stratification under global warming may critically affect future STYs.

Although several studies have indicated potential impacts on typhoon intensities from changes in ocean thermal structure under global warming[16–19], the effects of changing salinity stratification on typhoons remains unexplored. On average, the northwestern tropical Pacific receives some of the highest rainfall in the global tropics and hence is characterized by low-salinity surface waters. Furthermore, mounting evidence suggests that rainfall in the western tropical Pacific is increasing relative to evaporation[20,21] and the surface is freshening[22,23], leading to a strengthening of the upper-ocean salinity stratification. Hence, an assessment of the impacts of these salinity changes on future STYs is desirable. In this study, we address this issue using a combination of observations, coupled climate model simulations and the framework of Dynamic Potential Intensity (DPI), a variant of the thermodynamic air–sea Potential Intensity that accounts for ocean stratification effects[24]. We first show that during the period 1961–2008, the increase in DPI for STYs due to a freshening of the upper ocean is 53% stronger than the decrease in DPI caused by changes in temperature over the same period. Furthermore, we demonstrate that under twenty-first century climate model projections, the positive impacts on STY intensification from a continued freshening of the surface ocean negates about half of the suppressive effects from ocean thermal structure changes.

## Results

**Significance of salinity stratification for STY intensification.** We begin by examining the role of salinity stratification in the intensification of typhoons using 35 years (1979–2013) of typhoon track data, monthly mean vertical profiles of oceanic temperature and salinity from the Hadley Center's EN4 data set, and monthly mean vertical profiles of atmospheric temperature and relative humidity from ERA-Interim reanalysis to estimate DPI. The DPI is calculated with and without salinity stratification

(see Methods) at each typhoon track location over the region between 130°E–150°E and 5°N–30°N. At each location, we also compute the typhoon intensification tendency. Using all track locations, it is found that the DPI and intensification tendency are correlated at 0.55, regardless of whether salinity stratification is included. When we consider only those track locations where typhoons attain STY strength (maximum wind speed higher than $65 \, \mathrm{m \, s^{-1}}$), the correlation between DPI and typhoon intensification tendency is 0.42 when salinity is included and 0.38 when salinity is not included. Further, when we consider STYs of Category-5 strength (maximum wind speed higher than $70 \, \mathrm{m \, s^{-1}}$), the correlations between DPI and typhoon intensification tendency with and without salinity are 0.41 and 0.33, respectively. Thus, salinity contributes to nearly a third of the total variance in the intensification of the strongest typhoons. The correlations between DPI and intensification tendency for STYs, with and without salinity, are statistically different based on a z-test. This also suggests that although salinity stratification contributes substantially to the intensification of STYs, it may not have a significant impact on the intensification of weaker storms. To understand this further, we computed the cold wakes induced by typhoons at all track locations (see Methods). The mean SST cooling when salinity is included is 0.2 °C weaker compared with the cold wake when salinity is not included (Supplementary Table 1). On the other hand, the mean SST cooling for STYs is 0.4 °C weaker when salinity is included compared with when it is not. Hence, the impact of salinity stratification on SST cooling increases considerably for strong storms. Consequently, although the inclusion of salinity increases the DPI for STYs by nearly $5 \, \mathrm{m \, s^{-1}}$, when storm locations of all strength are considered, salinity enhances the DPI by only $2.3 \, \mathrm{m \, s^{-1}}$ (Supplementary Table 1). Thus, for stronger storms, the mixing generally extends deeper, and the likelihood that the typhoon's cold wake and intensification tendency will be influenced by upper-ocean salinity stratification is higher.

**Observed effect of salinity changes on STY intensities.** We now consider observed changes in sea surface salinity and their impacts on STYs over the 56-year historical period 1958–2013 using the EN4 data set. Trends in sea surface salinity, averaged over the major typhoon season of June–November, show that the surface ocean freshened over this period (Fig. 1a). A decrease in surface salinity occurred over almost the entire domain, with significant negative trends in the region between 120°E–160°E and 5°N–30°N. Surface salinity also decreased over a small region in the South China Sea between 110°E–120°E and 15°N–20°N. These trends in surface salinity are consistent with those shown in previous studies[22,23] and appear to be a part of longer-term trends due to an accelerating hydrological cycle driven by global warming[23]. There is a significant overlap between regions of strong surface freshening and high STY track density, indicating that changes in near-surface salinity have the potential to impact the intensity of STYs. Although ocean dynamics may play a role in the redistribution of near-surface waters[22], a strong correlation between surface salinity and satellite-derived precipitation suggests that increasing rainfall over the northwestern tropical Pacific might be a significant factor behind the freshening of the surface ocean in that region. The typhoon season-mean surface salinity is correlated with April–September mean rainfall from Global Precipitation Climatology Project at −0.6, for 1979–2013, highlighting the dominant control of precipitation over sea surface salinity in this region. Subsurface salinity trends, shown in Fig. 1b, lend support to this idea. Averaged over the region between 130°E–150°E and 5°N–25°N, they show that the strongest freshening occurs in the upper 50 m at a rate of ∼0.03 psu per decade. Similar results are obtained when salinity

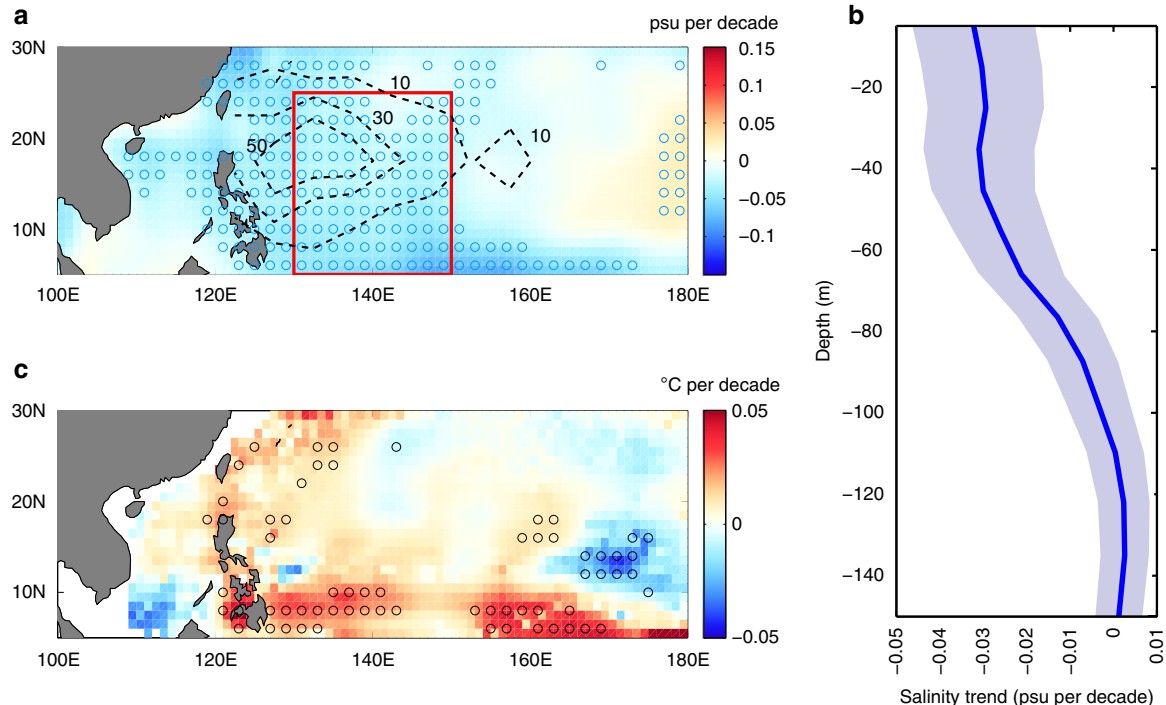

**Figure 1 | Observed trends in upper-ocean salinity and salinity impact on super typhoon cold wakes for 1958–2013 based on EN4.** (**a**) Trends in typhoon-season (June–November) mean sea surface salinity in psu per decade with dashed-black contours, at intervals of 20 units, overlaid representing the number of 6 hourly super typhoon (STY) track locations. The box in red indicates the study region: 130°E–150°E and 5°N–25°N. (**b**) Trends in typhoon-season mean subsurface salinity, averaged over the region 130°E–150°E and 5°N–25°N (see box in **a**), in psu per decade. The line represents the median trend value and the shaded area indicates the 95% confidence intervals, estimated using a linear regression. (**c**) Trends in typhoon-season (June–Nov) mean salinity contribution to cold wakes in °C per decade. In **a**,**c**, locations with statistically significant trends are marked with circles.

data based on the Simple Ocean Data Assimilation (SODA) reanalysis are used instead (Supplementary Fig. 1a,b).

Next, we examine the impact of these salinity changes on the intensity of STYs. The spatial pattern of salinity's contribution to trends in STY cold wakes (see Methods), shown in Fig. 1c, is consistent with that in surface salinity trends (Fig. 1a). Regions of decreasing trends in surface salinity are associated with positive trends in cold wakes (that is, reduced typhoon-induced cooling). This is because a reduction in surface salinity enhances ocean stratification, which in turn reduces the vertical mixing induced by a storm and, consequently the SST cooling. Figure 2a,b show time series of the contribution of salinity and temperature, respectively, to the DPI for STYs, averaged over the typhoon season and over the region 130°E–150°E and 5°N–25°N. To ensure the robustness of our results based on EN4, we perform similar analyses using three different ocean reanalysis products over a similar period (see Methods) and examine the ensemble mean time series and trends over the 48-year common period of 1961–2008. The mean trend in DPI due to salinity (Fig. 2a) is $0.196 \, \text{m s}^{-1}$ per decade and the mean trend in DPI due to changes in temperature alone is nearly $-0.3 \, \text{m s}^{-1}$ per decade (Fig. 2b). The trends from the individual reanalyses for both salinity (EN4: $0.116 \, \text{m s}^{-1}$ per decade, SODA: $0.174 \, \text{m s}^{-1}$ per decade, ORAS4: $0.120 \, \text{m s}^{-1}$ per decade and GFDL: $0.378 \, \text{m s}^{-1}$ per decade) and temperature (EN4: $-0.418 \, \text{m s}^{-1}$ per decade, SODA: $-0.144 \, \text{m s}^{-1}$ per decade, ORAS4: $0.000 \, \text{m s}^{-1}$ per decade and GFDL: $-0.614 \, \text{m s}^{-1}$ per decade) are broadly consistent. Although the mean trend in DPI due to salinity is statistically significant at the 99% level, the mean trend in DPI due to temperature is not significant.

To understand this further, we performed a multivariate regression analysis with the typhoon season-mean Niño 3.4 and the Pacific Decadal Oscillation indices as predictors and the time series of DPI due to temperature as the response. El Niño Southern Oscillation and Pacific Decadal Oscillation are the dominant modes of climate variability in the Pacific[25]. Nearly 59% of the variance in the temperature contribution to DPI is explained by the regression model, suggesting that the time series of DPI due to temperature is characterized by substantial interannual and decadal variability. When we remove the influence of these two major climate modes by subtracting the linear fit from the raw time series, the mean trend in the residual time series is $-0.128 \, \text{m s}^{-1}$ per decade but still not statistically significant. To understand the cause of the negative trend in cold wakes due to temperature changes, Fig. 2c shows trends in area- and typhoon-season averaged subsurface temperature. The magnitude of the positive trend decreases with depth. Thus, the larger increase in temperature near the surface relative to the subsurface, confirmed by analysis based on SODA oceanic temperature profiles (Supplementary Fig. 1c), causes the negative trend in DPI due to temperature. A similar regression analysis performed for the salinity contribution to DPI reveals that natural modes of climate variability have almost no impact on the trend in DPI due to salinity. Thus, when natural climate variability is accounted for, the mean trend in DPI due to salinity is 53% stronger than the mean trend in DPI due to temperature. The positive trend in DPI due to salinity reflects the importance of salinity for cold SST wakes of STYs.

**Future projections of salinity impacts on STY intensities.** So far, we have examined changes in salinity stratification and its potential impacts on the intensification of STYs over a nearly 50-year historical period. Next, we ask ourselves the following

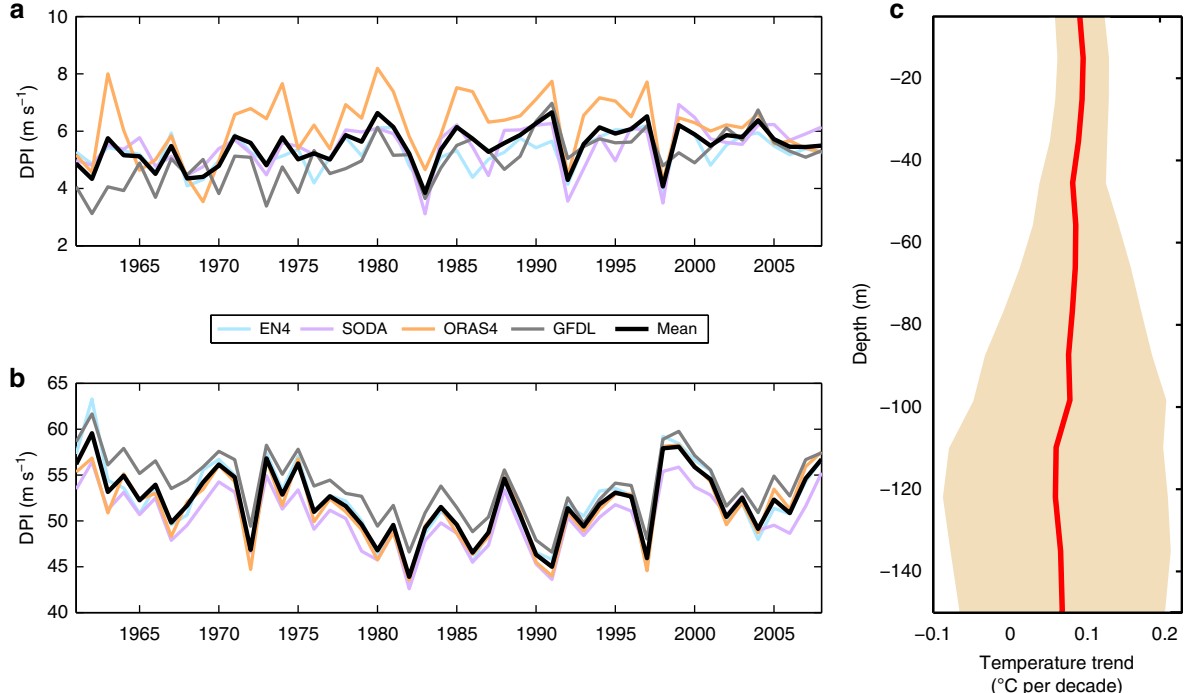

**Figure 2 | Observed trends in salinity and temperature impacts on super typhoon intensification.** Time series of (**a**) salinity and (**b**) temperature contribution to typhoon-season mean Dynamic Potential Intensity (DPI) for super typhoons (STY), averaged over the region 130°E–150°E and 5°N–25°N (see box in Fig. 1a), in ms$^{-1}$ per decade for various ocean data products (EN4, light blue; SODA, purple; ORAS4, orange; GFDL, grey) and the mean of all products (black). (**c**)Trends in typhoon-season mean subsurface temperature, averaged over the region 130°E–150°E and 5°N–25°N (see box in Fig. 1a), in °C per decade based on EN4 data. The line represents the median trend value and the shaded area indicates the 95% confidence intervals, estimated using a linear regression.

questions: How may the upper-ocean salinity stratification in the northwestern tropical Pacific evolve under global warming, and what are its impacts on the intensity of future STYs? To address these we analysed output from 18 different coupled climate models from Coupled Model Intercomparison Project—Phase 5 (CMIP5) under the Representative Concentration Pathway (RCP) 8.5 climate change scenario (see Methods). The 100-year changes in surface salinity and precipitation, based on an ensemble mean of the various models and computed as the difference between the means over 2081–2100 and 1981–2000 (Fig. 3a), indicate a continued strengthening of the hydrological cycle and near-surface salinity stratification through the twenty-first century. The largest decrease in the typhoon-season mean surface salinity, exceeding 0.1 psu per decade, occurs in the region to the east of the Philippines between 130°E–160°E and 5°N–15°N, where precipitation increases substantially. The change in upper-ocean salinity causes a strong decrease in cold wake magnitude (Fig. 3b). Averaged over the region 130°E–150°E and 5°N–25°N, the reduction in cold wake magnitude is ~0.01 °C per decade, a value significant at the 99% level based on a Student's *t*-test for difference of means. The freshening of the surface ocean enhances the upper-ocean salinity stratification and, consequently, reduces the vertical mixing and SST cooling.

To decipher the impact of projected salinity changes on STY intensities, we constructed probability distribution functions (PDFs) of climatological typhoon season-mean DPI values based on the multi-model ensemble mean. Each PDF represents the spatial distribution of DPI values over the region 130°E–150°E and 5°N–25°N. PDFs with and without salinity and their differences are shown (Fig. 3c,d). For the PDFs of DPI without salinity (Fig. 3d), we find that the occurrence of the highest intensities decreases by ~9% under climate change, consistent

with a decrease in DPI of 0.185 m s$^{-1}$ per decade averaged in the same region. Thus, changes in temperature under the RCP 8.5 scenario tend to reduce the mean DPI over the region 130°E–150°E and 5°N–25°N. To understand this further, we examine changes in sub-surface temperature. Maximum warming occurs at the surface and the warming magnitude decreases with depth (Supplementary Fig. 2a). These changes tend to magnify the storm-induced SST cooling that subsequently enhances the negative feedback on the intensity of future typhoons[18,19].

For the PDFs of DPI with salinity included (Fig. 3c), there are considerably more locations with DPI values exceeding 65 m s$^{-1}$ and fewer locations with DPI values below 55 m s$^{-1}$, compared with the DPI distribution without salinity (Fig. 3d). These DPI distributions are in agreement with results obtained earlier, which show that the mean DPI is higher for STYs when salinity is included. For the PDFs of DPI with salinity (Fig. 3c), we find that there are no significant changes in any of the bin sizes, suggesting that changes in salinity tend to negate the weakening impacts of temperature changes on DPI under the RCP 8.5 scenario. The increased freshening of the upper-ocean under climate change enhances the near-surface salinity stratification (Supplementary Fig. 2b) and reduces typhoon-induced SST cooling, a process that tends to invigorate typhoons. The net effect of salinity is to increase the area-averaged DPI by 0.118 m s$^{-1}$ per decade, in contrast to a decrease in DPI of 0.185 m s$^{-1}$ per decade due to temperature alone that was discussed earlier. Although the change in DPI due to salinity is significant at the 99% level across all models, based on the Student's *t*-test, the change in DPI due to temperature is not statistically significant.

Although these results were based on an ensemble mean of the various CMIP5 models used in our analysis, an examination of

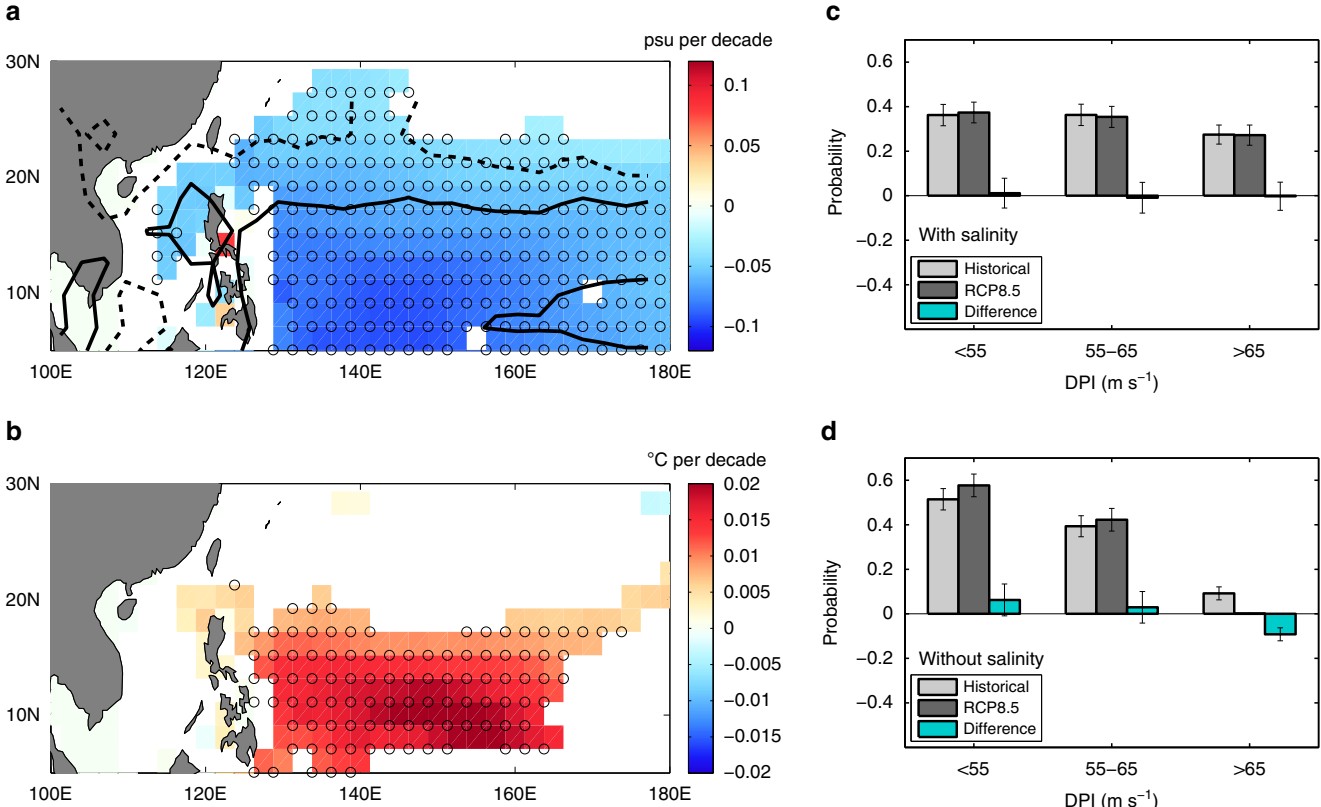

**Figure 3 | Projected 100-year changes in salinity impacts on super typhoon intensification based on the CMIP5 multi-model ensemble mean.**
(**a**) Changes in sea surface salinity (psu per decade) with contours of precipitation changes overlaid: the solid contour represents a precipitation change of 0.1 mm per day per decade, whereas the dashed contour represents a precipitation change of 0.05 mm per day per decade. (**b**) Contribution of salinity to changes in super typhoon (STY) cold wakes (°C per decade). In **a**,**b**, changes shown are statistically significant at the 95% level and locations where more than 15 out of 18 models agree on the sign of the change are marked in circles. PDFs of typhoon-season mean Dynamic Potential Intensity (DPI) distribution in ms$^{-1}$, (**c**) with and (**d**) without salinity, for STYs and for the region between 130°E–150°E and 5°N–25°N (see box in Fig. 1a). The PDFs for the historical period, based on years 1981–2000, are given in light grey. The PDFs for the future, based on years 2081–2100 under the RCP 8.5 scenario, are given in dark grey. The difference between the PDFs with and without salinity is given in cyan. For the PDFs shown in **c**,**d**, each bin size and the corresponding error bar, represented by ± one s.d., were computed based on the Monte Carlo method of repeated random sampling (see Methods).

the inter-model spread indicates that the salinity effects on DPI are consistent across models. The inter-model spread in the surface salinity signal is depicted in Fig. 4a. The ensemble-mean change in surface salinity projected by these models is −0.067 psu per decade, a value statistically significant at >99% based on a Student's t-test for difference of means. This confirms the robustness of the surface salinity signal in the northwestern tropical Pacific under climate change. Changes in DPI due to salinity from each of the 18 CMIP5 models are shown in Fig. 4b. In every model, the projected change in DPI due to salinity is positive. Examining the contributions from individual models, we note that the change in DPI due to salinity based on MIROC-ESM is unusually high (0.47 m s$^{-1}$ per decade) and is an outlier (see Methods). To understand the degree to which the contribution from this model influences the multi-model mean, we computed the ensemble mean change in DPI due to salinity without it. The mean change in DPI due to salinity from the other 17 models is 0.097 m s$^{-1}$ per decade, a value statistically significant at the 95% level. Therefore, the change in DPI due to salinity projected by the multi-model ensemble mean of CMIP5 for the northwestern tropical Pacific is a robust signal and is not contaminated by anomalies or outliers. On the other hand, the ensemble mean DPI change due to temperature is − 0.185 m s$^{-1}$ per decade (Fig. 4b). On examination, we find that even in the case of temperature, there are a few models whose contributions are unusually strong. For instance, the contribution

from MPI-ESM-LR is about − 0.5 m s$^{-1}$ per decade. However, none of the contributions from the models is an outlier in this case. Thus, salinity negates about half of the temperature effect on DPI.

## Discussion

In this study, we focused on the impacts of salinity changes on STYs of the northwestern tropical Pacific. However, to gain a broader perspective, we now briefly examine projected salinity changes in the Northern Hemisphere tropical cyclone basins and their possible impacts on DPI. In the Pacific, the freshening of the surface ocean extends all the way into the eastern Pacific (Fig. 5). Averaged over the central Pacific tropical cyclone basin and the western part of the eastern Pacific tropical cyclone basin (180°W–120°W, 5°N–25°N), the surface salinity reduces by − 0.038 psu per decade, a value significant at the 95% significance level. Consistent with this, the area-averaged DPI due to salinity increases by 0.115 m s$^{-1}$ per decade. This result is important from a societal standpoint given that the Hawaiian islands, located in this region, are prone to an enhanced threat from tropical cyclones under climate change[26]. Unlike the Pacific, in the North Atlantic the surface salinity is projected to increase, consistent with previous studies[23]. Averaged over the northwestern tropical Atlantic (70°W–40°W, 10°N–25°N), the surface salinity increases by 0.055 psu per decade, a value significant at

**a**

**b**

**Figure 4 | Inter-model spread in CMIP5 projections.** Changes in (**a**) Sea Surface Salinity (SSS) in psu per decade and (**b**) Dynamic Potential Intensity (DPI) due to salinity (grey) and due to temperature (white), in m s$^{-1}$ per decade, projected by 18 CMIP5 climate models under the RCP 8.5 scenario. SSS and DPI are averaged over the typhoon season (June–November) and over the region 130°E–150°E and 5°N–25°N (see box in Fig. 1a). In **a,b**, change is defined as the difference between the mean over the 20-year periods 2081–2100 and 1981–2000.

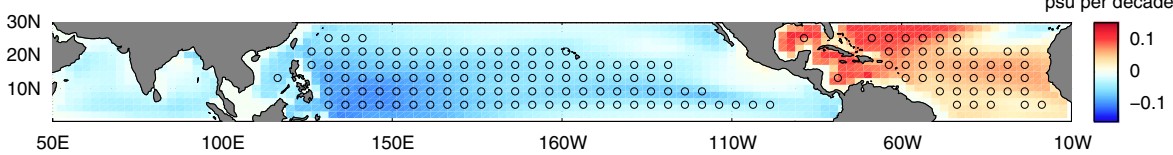

**Figure 5 | Projected 100-year changes in surface salinity.** Ensemble mean change in sea surface salinity (psu per decade), averaged over the typhoon season (June–November), under the RCP 8.5 scenario. Change is defined as the difference between the mean over the 20-year periods 2081–2100 and 1981–2000. Locations with statistically significant changes are marked with circles.

the 90% level. However, the DPI change due to salinity is not significant in the North Atlantic. Finally, in the Northern Indian Ocean, the changes in surface salinity are not statistically significant.

Recently, it has been suggested that the ocean subsurface may play a suppressive role in the intensification of future typhoons[18,19]. However, such studies considered only changes in the upper-ocean thermal structure and did not account for salinity. An increased tilting of the thermal profile can simultaneously have two competing effects: It can increase the ocean's negative feedback onto a typhoon, as mixing to a certain depth will produce more SST cooling and it can reduce the ocean's negative feedback because of enhanced stratification that causes a decrease in mixing-induced cooling. Which of the two effects ultimately prevails depends on the nature of the thermal profile and the strength of mixing. On the other hand, an increase in salinity stratification can only induce a positive feedback onto a storm and is thus less ambiguous. Here we show that in the

northwestern tropical Pacific, salinity stratification plays an important role in the intensification of STYs. In observations, the change in DPI due to salinity overwhelms the change in DPI due to temperature by a comfortable margin over the period 1961–2008. Under future projections using CMIP5 models, salinity negates ∼50–60% of the change in DPI due to temperature. Thus, our results suggest that changes in upper-ocean salinity induced by a changing hydrological cycle cannot be neglected when considering future typhoon activity. As part of efforts to improve our understanding of salinity's impacts on typhoon intensification, our study calls for an improved and sustained monitoring of the upper-ocean salinity structure in the northwestern tropical Pacific and its response to changes in the hydrological cycle as our planet warms.

## Methods

**Data.** EN4 monthly mean profiles of oceanic temperature and salinity[27] for the 56-year period 1958–2013 are obtained from UK Met Office's Hadley Center (http://www.metoffice.gov.uk/hadobs/en4/index.html) and used to examine trends in salinity and temperature, and to estimate cold wakes. NCEP-NCAR reanalysis[28] monthly mean atmospheric profiles of temperature and relative humidity, and monthly mean sea-level pressure, obtained for the same 56-year period from http://www.esrl.noaa.gov/psd/data/gridded/data.ncep.reanalysis.html, are used in tandem with EN4 data, to estimate DPI. Monthly mean profiles of oceanic temperature and salinity are also obtained from SODA 2.1.6 ocean reanalysis[29] (http://iridl.ldeo.columbia.edu/SOURCES/.CARTON-GIESE/.SODA/.v2p1p6/) for the period 1958–2008, from ECMWF's ORAS4 (ref. 30) (http://www.ecmwf.int/en/research/climate-reanalysis/ocean-reanalysis) for the period 1958–2013 and GFDL's ECDA 3.1 (ref. 31) (http://www.gfdl.noaa.gov/ocean-data-assimilation) for the period 1961–2010, and used with NCEP-NCAR atmospheric reanalysis data to estimate DPI, to ensure the robustness of our results based on EN4. Climatological monthly mean profiles of oceanic temperature and salinity are obtained from Argo[32] (http://apdrc.soest.hawaii.edu/projects/Argo/index.php) to perform numerical experiments with the Price–Weller–Pinkel (PWP) mixed layer ocean model. Typhoon track data for the period 1979–2013, obtained from the US Navy's Joint Typhoon Warning Center (http://www.usno.navy.mil/NOOC/nmfc-ph/RSS/jtwc/best_tracks/), are used to obtain track locations and typhoon intensities, and to estimate typhoon translation speeds and intensification tendencies. ERA-Interim monthly mean atmospheric profiles of temperature and relative humidity, and monthly mean sea-level pressure[33] are obtained for the 35-year period 1979–2013 from ECMWF (http://www.ecmwf.int/en/research/climate-reanalysis/era-interim) and used with EN4 data to estimate DPI for the shorter-term analysis with typhoon track data. For this analysis, we focus on the satellite period during which tropical cyclone data tends to be more reliable.

Time series of the Niño 3.4 and the PDO indices were obtained from http://www.cpc.ncep.noaa.gov/data/indices/ and http://ds.data.jma.go.jp/tcc/tcc/products/elnino/decadal/pdo.html, respectively. Monthly mean Global Precipitation Climatology Project precipitation data[34], obtained from http://precip.gsfc.nasa.gov/ for the 35-year period 1979–2013, are used to examine the influence of rainfall on surface salinity in the northwestern tropical Pacific. Daily mean microwave optimally interpolated (OI) SST data for the 11-year period 1998–2008, produced by Remote Sensing Systems and available at http://www.remss.com/, are used to validate our method of estimating cold wakes from monthly mean data. Output from eighteen coupled climate models belonging to the IPCC's CMIP5 (ref. 35), obtained from http://pcmdi9.llnl.gov/, are used to examine future changes in salinity, STY cold wakes and DPI. The 18 models are ACCESS 1.3, BNU-ESM, CAN-ESM2, CMCC-CM, CMCC-CMS, CNRM-CM5, CSIRO-MK3-6.0, GFDL-CM3, GFDL-ESM2G, GFDL-ESM2M, HADGEM2-CC, HADGEM2-ES, IPSL-CM5A-LR, MIROC-ESM, MPI-ESM-LR, NCAR-CCSM4, NCAR-CESM-BGC and NCAR-CESM-CAM5. Data from two 20-year periods, the historical period of 1981–2000 and a future period of 2081–2100, are used to estimate 100-year changes. For the future period, the RCP 8.5 climate change scenario is used.

**Computing the salinity effect on DPI.** The intensification tendency for typhoons is estimated as the linear regression coefficient of the maximum wind speed at six successive 6 hourly storm locations[12]. At each typhoon track location, the depth of typhoon-induced mixing (L) is estimated following a turbulent kinetic energy approach[24], based on a balance between the work done by the typhoon's wind forcing and the potential energy barrier created by ocean stratification as follows.

$$L = h + \left( \frac{2\rho_o \mathbf{u}_*^3 t}{\kappa g \alpha(T,S)} \right)^{\frac{1}{3}} \qquad (1)$$

Parameters used are the initial mixed layer depth $h^{36}$, the typhoon's maximum wind speed (to compute the friction velocity $\mathbf{u}_*$) and translation time ($t$), the von Kármán constant ($\kappa$), the acceleration due to gravity ($g$) and the rate of density

increase beneath the mixed layer $\alpha(T,S)$, which is a function of both oceanic temperature ($T$), as well as salinity ($S$). Thus, the actual SST experienced by the storm, $T_{dy}$, is given as the temperature averaged from the surface to the mixing depth ($L$). The magnitude of the cold SST wake induced by the typhoon at that location is then estimated as the difference between $T_{dy}$ and the initial SST. This method of estimating cold SST wakes using monthly mean oceanic temperature and salinity profiles is shown to perform reasonably well (see Supplementary Notes 1 and 2, Supplementary Figs 3 and 4, and Supplementary Table 2). To isolate the influence of temperature on cold wakes, we set salinity to zero and estimate the $T_{dy}$ using temperature only. The impact of salinity on cold wakes is then given as the difference between the cold wake estimated using the full $T_{dy}$ and the cold wake estimated using temperature-only $T_{dy}$.

The DPI is computed as

$$DPI^2 = \frac{T_{dy} - T_o}{T_o} \frac{C_{\mathbf{K}}}{C_{\mathbf{D}}} \left( K_{T_{dy}} - K \right) \qquad (2)$$

where DPI is expressed as the maximum wind speed of the typhoon, $T_o$ is the outflow temperature, $C_{\mathbf{D}}$ is the coefficient of drag, $C_{\mathbf{K}}$ is the coefficient of enthalpy, $K_{T_{dy}}$ is the enthalpy of air in contact with the sea surface and $K$ is the specific enthalpy of air near the surface in the ambient boundary layer. The impact of subsurface temperature on DPI is estimated as the DPI computed using temperature-only $T_{dy}$. The salinity influence on DPI is estimated as the difference between the DPI computed using the full $T_{dy}$ and the DPI computed using temperature-only $T_{dy}$. For the analysis using actual typhoon track data, we use information of the storm state at each track location to estimate the mixing depth. The cold wakes and DPI are averaged over a $4° \times 4°$ box centred over the eye of the storm to account for asymmetry in typhoon-induced SST cooling. For the climate scale analysis where individual storm states are not known, we use a typical storm translation speed of $5 \, \text{m} \, \text{s}^{-1}$ and a STY wind speed of $65 \, \text{m} \, \text{s}^{-1}$ to estimate the mixing depth.

**Estimating PDFs and outliers.** The DPI PDFs are estimated using the Monte Carlo method of repeated random sampling. We approximately select half the number of points in the sample set randomly and generate a PDF. We then repeat this process 10,000 times. The mean and s.d. for each bin, calculated across the PDFs, give the corresponding mean bin size and error. Outliers in CMIP5 model projections are detected using the 'median of absolute deviation' method[37], a technique that has been previously employed in geophysical studies.

**Numerical experiments to validate the DPI framework.** To assess the accuracy with which our formula predicts the salinity impact on STY mixing lengths and cold wakes, we performed numerical experiments using the PWP one-dimensional mixed layer ocean model[38]. In the PWP model, vertical mixing occurs when a critical bulk Richardson number criterion is satisfied. On the other hand, the mixing length in our formula is estimated using an energy-balance approach following the Monin–Obukhov theory[39]. Thus, a comparison with results from the PWP model serves as an independent test of our formulation.

We conducted 300 experiments with the PWP model, using Argo temperature and salinity profiles from the western Pacific (130°E–150°E and 5°N–30°N). Our methodology is as follows. For June through November, we obtain monthly gridded Argo climatological temperature and salinity profiles at a regular $4°$-lat. $\times 4°$-lon. grid in our study region, giving 25 sets of profiles for each of 6 months, or 150 total. We use each of these sets of profiles as initial conditions for a simulation of the PWP model. A time step of 15 min and vertical resolution of 1 m is used for the simulations[24]. The wind stress is calculated based on a variable drag coefficient proposed for high wind speeds[40]. In each experiment, we force the model with wind stress corresponding to a STY (maximum wind of $65 \, \text{m} \, \text{s}^{-1}$) that is translating northward at $5 \, \text{m} \, \text{s}^{-1}$, a radius of maximum winds of 50 km and a total storm radius of 200 km. As we are focusing on STYs, we use a single maximum wind speed. A meridional transect through the centre of the storm is obtained based on these parameters and under the assumption of an axisymmetric wind distribution[41]. We integrate the model from the outer radius of the typhoon, at which wind stress is zero, to the radius of maximum wind on the northern side of the storm, as the cold wake generated by these winds has the strongest influence on the storm's intensity and it is this cooling that our formula is designed to represent.

Two sets of simulations are run: one using the observed temperature and salinity as initial conditions (PWPfull) and one using observed temperature but salinity set to zero at every depth (PWPnosal). The goals of the simulations are to test our formula's ability to generate a realistic mixing length and cold SST wake and the ability to give realistic sensitivities of the mixing length and cold wake to salinity. We therefore compare mean values of these parameters, averaged over all months (June through November) and all locations in our study region. The model's cold wake is calculated as the SST on the final time step of integration minus the initial SST. The impact of salinity in our formula is determined by subtracting our formula's mixing length, calculated using the full temperature and salinity profiles for $\alpha(T,S)$ and mixed layer depth, from the mixing length calculated using the full temperature but setting salinity to zero at every depth (temperature based mixed layer depth in this case)[36]. The impact of salinity in PWP is calculated as the difference between the PWPfull and PWPnosal runs. The experiments reveal that the mixing lengths and SST cold wakes from our model agree reasonably well

with those from the PWP simulations. Please see Supplementary Note 1, Supplementary Fig. 3 and Supplementary Table 2 for further details.

**Code availability.** The programme to compute Potential Intensity is available at http://eaps4.mit.edu/faculty/Emanuel/products.

**Data availability.** The data that support the findings of this study are freely available to download from the URLs provided in Methods, or from the corresponding author upon request.

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

## Acknowledgements

This research was supported by the Office of Science of the U.S. Department of Energy (DOE) as part of the Regional and Global Climate Modeling programme. The Pacific Northwest National Laboratory is operated for DOE by Battelle Memorial Institute under contract DE-AC05-76RL01830. G.F. was funded by base funds to NOAA/AOML's Physical Oceanography Division. We thank the editor and three anonymous reviewers whose comments improved the quality of our manuscript significantly.

## Author contributions

K.B., G.R.F., L.R.L. and K.A.E. conceived the main idea and developed it. K.B. performed the analysis of observations and climate models. G.R.F. conducted numerical experiments. K.B. and G.R.F. wrote the paper with inputs from L.R.L. and K.A.E.

## Additional information

**Competing financial interests:** The authors declare no competing financial interests.

