## [Peer Review File · Nature Communications]

Reviewers' comments:

Reviewer #1 (Remarks to the Author):

Review of "Accelerating hydrological cycle, upper-ocean freshening and intensification of super typhoons" by Balaguru et al. submitted to Nature Communications.

Overview:

The authors use a predefined index-DPI to investigate the effect of upper-ocean salinity changes on the potential intensity of super typhoons. They illustrate that the upper-ocean has been freshened in the last decades based on multiple datasets and would be further freshened in future projected by nine CMIP5 models with increasing global temperature and intensifying global hydrological circulation. Based on the predefined DPI, the degree of upper-ocean freshening will pronouncedly decrease the super typhoon-induced ocean cooling and thus enhance super typhoons in future. The authors declare that the effect of upper-ocean freshening is as large as that of the changes in ocean temporal stratification.

Major comments:

I understand the upper-ocean is freshening under global warming and it should decrease the typhoon-induced ocean cooling and enhance future typhoon intensity. However, I highly doubt the sensitivity of TC-induced ocean cooling effect (the T_{dy}) as well as the DPI to the sea surface salinity (SSS) is as large as the authors declare. The present study does not show the details of calculating DPI and T_{dy} , but cites a paper Balaguru et al. 2015 GRL (hereafter to B15GRL). The key in DPI reflecting the contribution of SSS is the mixing length L , which is from an empirical formula. I don't find SSS in the Eq. 1 to calculate L in B15GRL. Is it included in sea surface density? B15GRL roughly compares the mixing lengths from the PWP model and predicted by T_{dy} formula. The results show that the T_{dy} - L is generally larger than the PWP- L . (Although the authors declare the overestimation of T_{dy} - L to the PWP- L is around 20%, a large portion of T_{dy} - L in B15GRL-Fig.1, especially the green and blue marks are larger than the corresponding PWP- L s around 50%.) The MOST IMPORTANT point is the sensitivity of T_{dy} - L to SSS is not estimated nor compared with the simulation of PWP models in B15GRL and the present study, which is the foundation stone of the present conclusion. Especially, how about the sensitivity of T_{dy} - L to SSS when ocean is stirred by super typhoon? I am also curious about the trend of the mixing length L in T_{dy} , which is a key node but not shown. Thus, I doubt the sensitivity of T_{dy} - L /DPI to SSS changes is overestimated.

Therefore, I recommend that sensitivity of ocean cooling effect in DPI to SSS changes must be evaluated, before the contribution of SSS changes to TC intensity is studied based on the DPI. For example, the sensitivity can be compared with the results using PWP model, even the 3D PWP model. The effect of SSS changes on ocean cooling can be also estimated in 3D PWP model with a spectrum of TC conditions, such as the moving speed and intensity, similar to the analyses done in Huang et al 2015 Nature Communications.

To study the future changes, the number of CMIP5 models used here, nine, is quite small. Actually, the rainfall changes differ greatly among the models, implying a large intermodel difference in SSS changes. Therefore, a larger group of model members could give a more reliable projection.

Reviewer #2 (Remarks to the Author):

General comment:

This paper addresses the impact of decreasing tendency of the surface salinity on super typhoons (STYs) in the western North Pacific. Using observed data and future projections by coupled models, the authors calculate Dynamic Potential Intensity (DPI) to demonstrate the impact of the upper ocean salinity. As they mentioned, ocean temperature structure change on tropical structure has been studied. On the other hand, the present study focuses on the importance of the salinity structure in the ocean. This is a new point and interesting for study of typhoon intensity. The reviewer considers that the findings of the paper are interesting and publication of the paper is expected. However, the reviewer has some concerns about some points in the manuscript. The authors are requested to revise the manuscript with considering the following comments.

Specific comments:

(P: page, Lt: line from the top of each page, Lb: line from the bottom)

1. P2, Lb8 and P11, Lt8: In this study, the vertical mixing in the upper ocean is considered for the SST reduction. On the other hand, strong winds associated with STYs reduce SST by large amounts of latent and sensible heat fluxes from the sea to the atmosphere. STYs also cause Ekman upwelling below the typhoon center or excite internal waves in the ocean (Price, 1981, JPO). These grid-scale motions also reduce the upper ocean temperature. The SST changes due to the heat fluxes and the grid-scale motions seem to be ignored in the present study. The definition of DPI on page 11 does not include these effects. Why are they not considered? The reduced SST (T_{dy}) in the equation of DPI is calculated by averaging temperature from the surface to the mixing depth (P11, Lt8). T_{dy} is also decreased by the above processes and then DPI is reduced. This means that the reduction of SST might be underestimated.
2. P13, Lt 11: The year 1982 of the reference 6 (Price, JPO) may be 1981.
3. P11, Lb2: The equation of DPI is essentially the same as Eq. (1) of Emanuel (1999, Nature). The SST of the Eq. (1) is replaced by T_{dy} in the present study. The effect of salinity is solely included in T_{dy} . Showing the equation of T_{dy} as a function of salinity could be helpful for readers to understand the relationship between salinity and DPI.
4. Title: The title of the paper is not appropriate. Because the hydrological cycle is not a main research topic of the present study and is only used as a context of "upper-ocean freshening" owing to rainfall.
5. P4, Lt 4 and Fig. 1A: The authors compare the trends of the sea-surface salinity (SSS) and number of STYs in Fig. 1A and conclude that the changes of SSS have impact to the STY intensity. This seems to be logically incorrect. The trends should be compared to a trend of STY but not to the number or density of STY. In Fig. 1A, the definition of the density of STY is unclear. Please indicate the unit of dashed lines in Fig. 1A.
6. P4, Lt 8: The authors insist that increasing rainfall (P) might cause freshening of the ocean surface. However, evaporation (E) is another factor to change SSS. The difference P-E might be more direct factor to change SSS. It is suggested to show a trend of P-E or a trend of P in Fig. 1 as in Fig. 3A?
7. P6, Lb 4: As same in the above comment, the distribution of trend of P-E might be better than that of P in Fig. 3A.
8. P8, Lt 7: The authors concluded that SSS changes are strong enough to cancel out the negative effects of temperature change. It is not clear what the negative effects are.
9. Figure 1: The positive trends in cold wakes are the largest to the south of 10N, where STYs

density is very small and typhoons are relatively weak. This might indicate that the decrease of SSS is not correlated with intensification of STYs.

10. Supplementary Information Section 1: In this section, DPI is compared to the intensification tendency of typhoons. DPI is an intensity of typhoon while the latter is time-derivative of intensity. The intensification tendency which is speed of intensification of typhoon is determined by complex development processes of typhoon. It is not clear why the time-derivative of intensity is compared to the potential intensity.

Reviewer #3 (Remarks to the Author):

A- The authors address an important question: how changes in ocean stratification (more specifically the part related to salinity) have affected and will affect the ocean feedback to typhoon intensities? Their results show that salinity changes are important to take into account in future projections of typhoon intensities. They provide evidence that future changes in salinity profiles could suppress the effect of future changes in temperature profiles in term of typhoon-induced cold wake and feedback to its intensity.

B- This is new and important result for the community.

C- I did not notice any grey zone in their methodology: they use a state-of-the art theory to estimate the ocean response to typhoons as well as several datasets to check the robustness of their results.

D- Simple statistical methods are used. This seems sufficient to support the results (apart from the use of CMIP5 multi-model mean, see F).

E- The conclusions are well supported by all the analyses presented in this paper.

F- Suggested improvements :

F1- In addition to the CMIP5 multi-model mean, it is important to show how much DPI increases from historical to RCP85 with and without salinity for each individual model. Indeed, there is no evidence here that the salinity trend is found in each model, and we could imagine that an outlier could produce a trend in the multi model mean that is found in no other individual model.

F2- All the datasets used in this paper have a global coverage, so the methodology could have been applied at a global scale without much more effort. Did the authors check other basins and did not find any significant influence of salinity? Jourdain et al. (JPO, 2013) reported contrasted ocean response to tropical cyclones in seven cyclonic regions due to differences in salinity stratification. Therefore, one could expect to find contrasted response to salinity changes in future emission scenarios between different cyclonic regions.

G- There is appropriate credit to previous work.

H- The manuscript is clearly written and sufficient methodological information is provided.

We would like to thank the editor and all three reviewers for their constructive and positive comments that helped improve our manuscript substantially. Besides addressing the concerns of the various reviewers, we corrected a small error in our code related to the analysis of observations, shown in Figure 2 of the main manuscript. The corrected results do not alter our conclusions significantly, but only make them stronger and suggest a dominant role played by salinity changes in DPI trends over the near 50-year observational period.

We thank reviewer #1 for the thoughtful comments. Here's our point-by-point response to your comments.

I understand the upper-ocean is freshening under global warming and it should decrease the typhoon-induced ocean cooling and enhance future typhoon intensity. However, I highly doubt the sensitivity of TC-induced ocean cooling effect (the Tdy) as well as the DPI to the sea surface salinity (SSS) is as large as the authors declare. The present study does not show the details of calculating DPI and Tdy, but cites a paper Balaguru et al. 2015 GRL (hereafter to B15GRL). The key in DPI reflecting the contribution of SSS is the mixing length L, which is from an empirical formula. I don't find SSS in the Eq. 1 to calculate L in B15GRL. Is it included in sea surface density? B15GRL roughly compares the mixing lengths from the PWP model and predicted by Tdy formula. The results show that the Tdy-L is generally larger than the PWP-L. (Although the authors declare the overestimation of Tdy-L to the PWP-L is around 20%, a large portion of Tdy-L in B15GRL-Fig.1, especially the green and blue marks are larger than the corresponding PWP-Ls around 50%.) The MOST IMPORTANT point is the sensitivity of Tdy-L to SSS is not estimated nor compared with the simulation of PWP models in B15GRL and the present study, which is the foundation stone of the present conclusion. Especially, how about the sensitivity of Tdy-L to SSS when ocean is stirred by super typhoon? I am also curious about the trend of the mixing length L in Tdy, which is a key node but not shown. Thus, I doubt the sensitivity of Tdy/L/DPI to SSS changes is overestimated.

Therefore, I recommend that sensitivity of ocean cooling effect in DPI to SSS changes must be evaluated, before the contribution of SSS changes to TC intensity is studied based on the DPI. For example, the sensitivity can be compared with the results using PWP model, even the 3D PWP model. The effect of SSS changes on ocean cooling can be also estimated in 3D PWP model with a spectrum of TC conditions, such as the moving speed and intensity, similar to the analyses done in Huang et al 2015 Nature Communications.

The sea surface salinity, as well as subsurface temperature and salinity, are included in the stratification parameter (α) in the mixing length formula. As detailed in Balaguru et al. (GRL, 2015; hereafter B15GRL), α is the linear rate of change of density below the mixed layer. We use density at the base of the mixed layer and at a depth of 50 m below the base of the mixed layer to calculate α . We have clarified this in the description of the mixing length equation in the manuscript. The reviewer is correct that, on average, the mixing length from our formula is about 20% larger than the mixing layer depth predicted by the PWP model, based on the analysis presented in B15GRL. However, that analysis included only three different density profiles from the western Pacific, corresponding to weak, medium, and strong stratification cases, in addition to three profiles from the eastern Pacific and three from the Atlantic, and it did not explicitly address the influence of salinity on the mixing length.

Following the reviewer's suggestion, we have therefore conducted 300 additional experiments with the PWP model, using temperature and salinity profiles from the western Pacific region that is used in our study (130°-150°E, 5°-25°N). Our methodology is as follows. For June through November, we obtain

monthly gridded Argo climatological temperature and salinity profiles at a regular $4^\circ\text{-lat.} \times 4^\circ\text{-lon.}$ grid in our study region, giving 25 sets of profiles for each of six months, or 150 total. We use each of these sets of profiles as initial conditions for a simulation of the PWP model. Following B15GRL, we use a time step of 15 minutes and vertical resolution of 1 m. Wind stress is calculated based on a variable drag coefficient, as described in the main text of the manuscript. For each PWP simulation, we force the model with wind stress corresponding to a super typhoon (STY) that is translating northward at 5 m/s with maximum wind of 65 m/s, radius of maximum wind of 50 km, and total storm radius of 200 km. Because we are focusing on STYs, we use a single maximum wind speed instead of a range of values as was done in B15GRL. A meridional transect through the center of the storm is obtained based on these parameters and under the assumption of an axisymmetric wind distribution and a radial dependence following DeMaria (1987) and B15GRL. We integrate the model from the outer radius of the typhoon, at which wind stress is zero, to the radius of maximum wind on the northern side of the storm, since the cold wake generated by these winds has the strongest influence on the storm's intensity, and it is this cooling that our formula is designed to represent. Two sets of simulations are run: one using the observed temperature and salinity as initial conditions (PWP_{full}), and one using observed temperature but salinity set to zero at every depth (PWP_{nosal}).

The goals of the simulations are to test our formula's ability to generate a realistic mixing length and cold SST wake, and the ability to give realistic sensitivities of the mixing length and cold wake to salinity. We therefore compare mean values of these parameters, averaged over all months (June through November) and all locations in our study region. The mixed layer depth in the PWP model, taken as a rough estimate of the mixing length, is calculated based on the criterion of an increase in density of 0.07 kg m^{-3} from the surface. The model's cold wake is calculated as the SST on the final time step of integration minus the initial SST. The impact of salinity in our formula is determined by subtracting our formula's mixing length, calculated using the full temperature and salinity profiles for α and mixed layer depth (a density criterion of 0.07 kg m^{-3} is used for the mixed layer depth), from the mixing length calculated using the full temperature but setting salinity to zero at every depth (mixed layer depth in this case is calculated using the temperature equivalent of a 0.07 kg m^{-3} increase in density, which is about 0.2°C). The impact of salinity in PWP is calculated as the difference between the PWP_{full} and PWP_{nosal} runs.

Consistent with the B15GRL analysis (their Figure 1), we find that on average our formula produces a mixing length that is about 23% larger than the final mixed layer depth (the value at the end of each simulation) produced by the PWP model (93.5 m from our formula, 76.0 m from PWP). The reduction in mixing length when salinity is included is likewise similar (14.4 m for our formula versus 11.1 m in PWP), and the percentage reduction is nearly the same (14.6% for PWP, 15.4% for our formula). These similarities occur despite differences in the representation of mixing in our formula, which is based on the Monin-Obukhov length, and the PWP model, which uses the bulk and gradient Richardson numbers. Our formula solves only for the mixing length, the depth to which mixing is confined, and not the full vertical profiles of temperature and salinity as in the PWP model (see Figure R1). For this reason, the final PWP mixed layer and our formula's mixing length do not agree perfectly. However, because the Monin-Obukhov mixing length is on the same order as the mixed layer depth (e.g., Cushman-Roisin 1994), the cold SST wakes are similar when calculated from the PWP model and our formula (-1.41°C for PWP compared to -1.24°C for our formula). The reductions in cold wake magnitude due to salinity are also comparable (0.38°C in our formula and 0.30°C in PWP). An additional factor contributing to the good agreement between the PWP model and our formula is the short-duration, high-intensity, wind-driven mixing associated with a STY, for which the more complicated depth- and time-dependent vertical mixing in the PWP model can be approximated well by the simpler energy balance employed by our formula.

In summary, there are reasonable agreements between the mixing lengths and SST cold wakes from our model and those from the PWP simulations. Similarly, the sensitivities of the mixing lengths and cold wakes to salinity are comparable. For all parameters, the agreement is within $\pm 27\%$. For further validation, we compared our formula's cold SST wakes to the cold wakes measured by satellite microwave sensors, based on the observed tracks of all STYs in our study region during 1998-2008 (Figure R3). We used a $4^\circ \times 4^\circ$ region centered on the storm to calculate the difference between the mean satellite SST three days before and one day after the storm's passage. The same region was used to create monthly mean temperature and salinity profiles from EN4 for calculation of the cold wake using our formula. We found good agreement between the mean cold wakes: -1.44°C for our formula and -1.52°C for satellite data.

We have added a table to the supplement summarizing the comparison between the PWP results and our formula. We also added Figure R1 (SI Figure 3 in the revised supplement) and Figure R3 (SI Figure 4 in the revised supplement).

Figure R1 (a) Magnitude of wind stress used to force the PWP model. (b) Evolution of ocean temperature during a typical PWP model run, using initial conditions from September and from the center of the western Pacific region used in the study. Triangles are mixed layer depth, calculated as the depth at which density is 0.07 kg m^{-3} greater than at the surface. The horizontal black line represents the mixing length from our formula.

To study the future changes, the number of CMIP5 models used here, nine, is quite small. Actually, the rainfall changes differ greatly among the models, implying a large intermodel difference in SSS changes. Therefore, a larger group of model members could give a more reliable projection.

Thank you for this comment. The

reviewer is correct that we used a moderate number of climate models in our study. However, the change in upper-ocean salinity under global warming is a robust signal that has been reported in earlier studies as well. Using a suite of observations, and simulations from coupled climate models belonging to CMIP3, Durack et al. (Science, 2012) showed that there is an amplification of the upper-ocean salinity pattern under global warming. Later, a similar analysis indicated a consistent response from CMIP5 models (Durack, Oceanography, 2015).

In our study, nine models were randomly selected from those used by Emanuel (PNAS, 2013) and Durack et al. (Science, 2012). However, we agree with the reviewer that using a larger number of models might enhance the confidence in our estimates. Thus, following the reviewer’s suggestion, we analyzed 9 more models from the CMIP5 archive, bringing the total number of models in our analysis to 18.

Figure R2 Changes in sea surface salinity, averaged over the typhoon season (June–November) and over the region 130E–150E, 5N–25N, projected by 18 CMIP5 climate models under the RCP 8.5 scenario. ‘Change’ is defined as the difference between the mean over the 20-year periods 2081–2100 and 1981–2000.

The inter-model spread in the surface salinity signal is depicted in Figure R2. The ensemble-mean change in surface salinity projected by these models is -0.067 (psu/decade), a value statistically significant at more than 99% based on a Student’s t-test for difference of means. This confirms the robustness of the surface salinity signal in the northwestern tropical Pacific under climate change. The ensemble-mean change in DPI due to salinity based on these 18 models is 0.118 m/s/decade. On the other hand, the change in DPI due to temperature is -0.185 m/s/decade. Thus, the change in DPI due to salinity cancels nearly 65% of the change in DPI due to temperature for STYs under global warming. These results are broadly consistent with the main conclusions reported in the previous version of our manuscript. We updated the main manuscript with these results and included Figure R2 as SI Figure 5 in the supplement.

We thank reviewer #2 for the thoughtful comments. Here's our point-by-point response to your comments.

P2, Lb8 and P11, Lt8: In this study, the vertical mixing in the upper ocean is considered for the SST reduction. On the other hand, strong winds associated with STYs reduce SST by large amounts of latent and sensible heat fluxes from the sea to the atmosphere. STYs also cause Ekman upwelling below the typhoon center or excite internal waves in the ocean (Price, 1981, JPO). These grid-scale motions also reduce the upper ocean temperature. The SST changes due to the heat fluxes and the grid-scale motions seem to be ignored in the present study. The definition of DPI on page 11 does not include these effects. Why are they not considered? The reduced SST (T_{dy}) in the equation of DPI is calculated by averaging temperature from the surface to the mixing depth (P11, Lt8). T_{dy} is also decreased by the above processes and then DPI is reduced. This means that the reduction of SST might be underestimated.

Thank you for bringing up this point. Tropical Cyclones (TCs) intensify by extracting heat energy from the upper ocean. Therefore, the critical environmental parameter that drives a TC's intensification is the thermal disequilibrium at the air-sea interface, making the SST felt by the core of the storm very important. Several processes play a role in determining the true SST felt by the storm, such as vertical mixing, loss of heat through fluxes, Ekman pumping, etc., and the relative significance of each of these processes has been extensively studied previously.

The sea surface under the TC is cooled primarily by two mechanisms: 1) TC-ocean heat exchange and 2) Vertical mixing of cooler sub-surface water (Price, J. F. *Ocean Sci.*, 5, 351–368, 2009). If heat exchange between the storm and the ocean were the dominant of mechanism, the upper ocean would cool only by a few tenths of a degree, based on the time integrated latent and sensible heat fluxes under a TC. However, both observations and models consistently indicate strong cooling of the upper ocean without necessarily a significant change in the upper-ocean heat content. This suggests that vertical mixing is most likely responsible for much of the SST cooling under the TC (Price, J. F. *Ocean Sci.*, 5, 351–368, 2009).

Several observational and modeling studies have found that vertical turbulent mixing is the dominant mechanism for the sea surface cooling under the storm. D' Asaro, E. A. et al. (*Geophys. Res. Lett.*, 34.15, 2007) measured the three dimensional structure of the upper-ocean response to Hurricane Frances and found that most of the SST cooling that occurred could be attributed to vertical mixing of cooler sub-surface water. Korty, R. L. (*25th Conference on Hurricanes and Tropical Meteorology*. 2002) performed a series of numerical experiments with a hurricane model coupled to a 3D ocean model and examined the significance of various processes in SST cooling induced by TCs. It was found that entrainment cooling due to vertical mixing played the dominant role in the SST response to a TC, with Ekman pumping and other non-local processes playing a relatively minor role. Even for storms moving very slowly, including non-local processes causes a difference in SST cooling of only about 5%. Besides these two, there are several other observational and modeling studies (for example: Jacob et al., *J. Phys. Oceanogr.*, 30, 1407–1429, 2000; Sanford et al., *Geophys. Res. Lett.*, 34, L13604, 2007) that reached similar conclusions regarding the role of vertical mixing in TC-induced cooling.

The overwhelming evidence supporting the role of upper-ocean vertical mixing as the most dominant process governing TC-induced SST cooling has prompted several studies (Price, J. F. *Ocean Sci.*, 5, 351–368, 2009; Lin, I-I. et al., *Geophys. Res. Lett.*, 40.9, 2013), including ours (Balaguru, K. et al., *Geophys. Res. Lett.*, 42.16, 2015), to modify metrics that are traditionally used to understand TC intensification in order to account for TC-induced vertical mixing. In this study, we use the framework

of Dynamic Potential Intensity (Balaguru, K. et al., *Geophys. Res. Lett.*, 42.16, 2015) in which the true SST felt by the storm (T_{dy}) is estimated dynamically using the prevailing storm state and ocean stratification conditions. The cold wakes are then computed as the difference between T_{dy} and the pre-storm SST. To validate our method of estimating cold wakes, we compared them with cold wakes estimated directly using satellite SST data (Figure R3, included as SI Figure 4 in the supplement). Our analysis reveals that the mean SST cooling induced by STYs is -1.44°C when estimated using our method and monthly mean oceanic temperature and salinity profiles from EN4. On the other hand, the mean SST cooling is -1.52°C when estimated directly using satellite SST data. The difference between the cold wakes estimated using these two methods is statistically insignificant based the Student's t-test for difference of means. Based on these results, our technique is reasonable and does not lead to significant underestimation of cold SST wakes.

Figure R3 SST cooling estimated using our technique (X-axis) plotted against SST cooling estimated directly using satellite microwave SST data (Y-axis).

P13, Lt 11: The year 1982 of the reference 6 (Price, JPO) may be 1981.

Thanks for noting this. The typo has been corrected.

P11, Lb2: The equation of DPI is essentially the same as Eq. (1) of Emanuel (1999, Nature). The SST of the Eq. (1) is replaced by T_{dy} in the present study. The effect of salinity is solely included in T_{dy} . Showing the equation of T_{dy} as a function of salinity could be helpful for readers to understand the relationship between salinity and DPI.

Thanks for pointing this out. We now indicate that the stratification term ‘ α ’ is a function of both temperature and salinity.

Title: The title of the paper is not appropriate. Because the hydrological cycle is not a main research topic of the present study and is only used as a context of "upper-ocean freshening" owing to rainfall.

Thanks for this comment. We modified our title to “Global warming-induced upper-ocean freshening and the intensification of super typhoons”.

P4, Lt 4 and Fig. 1A: The authors compare the trends of the sea-surface salinity (SSS) and number of STYs in Fig. 1A and conclude that the changes of SSS have impact to the STY intensity. This seems to be logically incorrect. The trends should be compared to a trend of STY but not to the number or density of STY. In Fig. 1A, the definition of the density of STY is unclear. Please indicate the unit of dashed lines in Fig. 1A.

Thanks for bringing up this point. We would like to clarify that we did not compare trends in SSS with the number of STYs in Figure 1A. Our only intention of showing the track density of STYs, in which the contours represent the number of 6-hourly STY locations, is to show that surface salinity has a negative trend in a region where STYs generally tend to form.

P4, Lt 8: The authors insist that increasing rainfall (P) might cause freshening of the ocean surface. However, evaporation (E) is another factor to change SSS. The difference P-E might be more direct factor to change SSS. It is suggested to show a trend of P-E or a trend of P in Fig. 1 as in Fig. 3A?

Thanks for bringing up this point. In our study, the observational analysis begins in 1958. However, direct precipitation measurements are only available from 1979 through GPCP. Hence, we are unable to show trends in precipitation for the period of observational analysis. But, we provide the correlation between precipitation and sea surface salinity over the western Pacific for the period 1979-2013 (-0.6), a value that is statistically significant at the 95% level and demonstrates the strong control of precipitation over surface salinity. Also, we cite previous studies (such as Durack, P. J. et al., 2012) which show that changes in surface salinity over the last 50 years indicate an amplification of the ‘precipitation minus evaporation (P-E)’ pattern, which is expected under global warming.

P6, Lb 4: As same in the above comment, the distribution of trend of P-E might be better than that of P in Fig. 3A.

Thanks for bringing this up. 100-year changes in precipitation, evaporation and their difference under the RCP8.5 scenario, based on the multi-model ensemble mean is shown in Figure R4. It is clear that changes in precipitation (P) dominate those in P-E and that the contribution of changes in evaporation (E) is relatively minor. Hence, our use of the pattern of precipitation changes to explain changes in surface salinity from CMIP5 models is reasonable.

Figure R4 Ensemble mean 100-year changes in a) Precipitation (P), b) Evaporation (E) and c) P-E, in mm/day, under the RCP 8.5 scenario and averaged over the typhoon season (June-November). Change is defined as the difference between the means over 2081-2100 and 1981-2000.

P8, Lt 7: The authors concluded that SSS changes are strong enough to cancel out the negative effects of temperature change. It is not clear what the negative effects are.

Thanks. By the negative effects of temperature change, we mean the reduction in the STY potential intensity caused by changes in temperature. This is now clarified.

Figure 1: The positive trends in cold wakes are the largest to the south of 10N, where STYs density is very small and typhoons are relatively weak. This might indicate that the decrease of

SSS is not correlated with intensification of STYs.

Thanks for noting this. The reviewer is right in pointing out that the strongest impact of salinity on STY cold wakes is to the south of 10N. However, salinity changes reduce cold wakes also in many areas where STYs tend to occur. The impact of changing salinity stratification is clear when we consider its impact on DPI (Figure 2A). Averaged over the region 130E-150E and 5N-25N, the DPI due to salinity shows a statistically significant positive trend, emphasizing the tendency of upper-ocean salinity changes to reduce STY cold wakes over that region.

Supplementary Information Section 1: In this section, DPI is compared to the intensification tendency of typhoons. DPI is an intensity of typhoon while the latter is time-derivative of intensity. The intensification tendency which is speed of intensification of typhoon is determined by complex development processes of typhoon. It is not clear why the time-derivative of intensity is compared to the potential intensity.

Thanks for bringing this up. The DPI is the theoretical limit to the maximum intensity that can be achieved by a TC under prevailing environmental conditions. Hence, for a TC with a given intensity, the higher (lower) the DPI, the larger (smaller) is its tendency to intensify. Thus, the DPI also imposes a constraint on TC intensification rates. The DPI has previously been used to understand TC intensification tendencies (Balaguru, K. et al., *Geophys. Res. Lett.*, 42.16, 2015).

We thank reviewer #3 for the thoughtful comments. Here's our point-by-point response to your comments.

In addition to the CMIP5 multi-model mean, it is important to show how much DPI increases from historical to RCP85 with and without salinity for each individual model. Indeed, there is no evidence here that the salinity trend is found in each model, and we could imagine that an outlier could produce a trend in the multi model mean that is found in no other individual model.

Thanks for bringing up this point. In our original paper, we reported the ensemble mean change in DPI due to salinity and temperature along with its statistical significance. The Student's t-test for difference of means examines the mean change normalized by the sample standard deviation. In other words, it gives a measure of the change relative to internal variability. Hence, if an outlier dominates the mean change, usually the t-test fails to produce statistical significance.

In response to this concern and reviewer #1's comment regarding the confidence in our estimates, we have increased the number of CMIP5 models in our analysis from 9 to 18. Please consider Figure R2, in which the surface salinity changes from each of the 18 models is shown. It is clear that the freshening of the surface ocean is consistent across all models. The ensemble mean 100-year change in surface salinity is -0.67 psu, a value significant at more than 99%.

Changes in DPI due to salinity from each of the 18 CMIP5 models are shown in Figure R5. In every model, the projected change in DPI due to salinity is positive. On an average, salinity changes induce a DPI change of 0.118 m/s/decade, a value significant at more than 99% level. Examining the contributions from individual models, we note that the change in DPI due to salinity based on MIROC-ESM is unusually high (0.47 m/s/decade). This is consistent with projected changes in surface salinity since the surface freshening projected by MIROC-ESM is the highest among the 18 models (Figure R2). To understand the degree to which the contribution from MIROC-ESM influences the multi-model mean, we computed the ensemble mean change in DPI due to salinity without MIROC-ESM. The mean change in DPI due to salinity from the other 17 models is nearly 0.1 m/s/decade, a value statistically significant at the 95% level based on the Student's t-test. Therefore, the change in DPI due to salinity projected by the multi-model ensemble mean of CMIP5 for the Northwestern tropical Pacific is a robust signal and is not contaminated by anomalies or outliers.

On the other hand, the ensemble mean DPI change due to temperature is -0.185 m/s/decade. Upon examination, we find that even in the case of temperature, there are a few models whose contributions are unusually strong. For instance, the contribution from MPI-ESM-LR is about -0.5 m/s/decade. When we compute the ensemble mean without that model, the multi-model mean projected changes in DPI due to temperature drops to about -0.167 m/s/decade. However, unlike salinity, the mean change in DPI due to temperature is not statistically significant, with or without the contribution from MPI-ESM-LR. Figure R5 is now shown and discussed in the supplement (SI Figure 6).

Figure R5 Changes in DPI due to salinity (grey) and due to temperature (white), averaged over the typhoon season (June-November) and over the region 130E-150E, 5N-25N, projected by 18 CMIP5 climate models under the RCP 8.5 scenario. ‘Change’ is defined as the difference between the mean over the 20-year periods 2081-2100 and 1981-2000.

All the datasets used in this paper have a global coverage, so the methodology could have been applied at a global scale without much more effort. Did the authors check other basins and did not find any significant influence of salinity? Jourdain et al. (JPO, 2013) reported contrasted ocean response to tropical cyclones in seven cyclonic regions due to differences in salinity stratification. Therefore, one could expect to find contrasted response to salinity changes in future emission scenarios between different cyclonic regions.

Thanks for this comment. We agree with the reviewer that looking at other regions, besides the western Pacific, would be interesting. Following the reviewer’s suggestion, we expanded our domain to all major TC basins of the Northern Hemisphere. Please consider Figure R6, which shows surface salinity changes based on the multi-model ensemble mean. Although there are sea surface salinity changes in other TC basins as well, the changes are statistically significant only in the western Pacific. Consistent with this, changes in DPI due to salinity are also significant in the western Pacific alone.

Figure R6 Ensemble mean change in sea surface salinity (psu), averaged over the Northern hemisphere TC season (June-November), under the RCP 8.5 scenario. ‘Change’ is defined as the difference between the mean over the 20-year periods 2081-2100 and 1981-2000. Stippling indicates those regions where the change is statistically significant at the 95% level based on the Student’s t-test.

Reviewers' comments:

Reviewer #1 (Remarks to the Author):

Review of "Global warming-induced upper-ocean freshening and the intensification of super typhoons" by Balaguru et al. submitted to Nature Communications.

The authors have clearly gone to considerable effort to respond to my previous concerns. I still have a major issue that I feel need addressing prior to publication.

Specific comment:

The most attractive conclusion in the original draft is that "the positive effect of this freshening is NEARLY STRONG ENOUGH TO CANCEL the suppressive effects of global warming-induced ocean thermal structure changes examined in previous studies." This conclusion is roughly acceptable, when the positive effect of the freshening is around 88.3% (1.43/1.62) of the suppressive effect of ocean thermal structure changes in the original draft. This conclusion was made based on only nine members in the original submission.

In the revision, the authors use a larger group of ensemble members, 18, to estimate the effect of upper-ocean freshening. However, the ratio of these two effects is only around 63.8% (1.18/1.85) in the revised draft, when more models are considered. The large decrease of the positive effect is due to the fact that there is an apparent outlier model in the original nine models, MIROC-ESM, with extremely high effect of SSS changes around 4.7. When more models are included, the extraordinary contribution of MIROC-ESM is reduced.

If we exclude this apparent outlier model, the mean of the other 17 models is only around 0.97, decreasing around 20%. The authors avoid this important issue in the main text. In the supplementary materials, the authors discuss this issue but state that there is also a model with a relatively large effect of ocean thermal structure changes. Of course, there must be one model possessing the largest value. If we exclude the model with the largest suppressive effect, the mean of the other models are around 1.67. Accordingly, the ratio of these two effects further decreases to 58% (0.97/1.67).

Another metrics for a group of values is the median of the values. The model with the ninth largest positive effect of the freshening among the 18 models is HadGEM-CC with a positive effect around 1.07, whereas the model with the ninth largest suppressive effect of is CMCC-CM with a negative effect around 2.0 [The values are roughly measured from Fig. S6 by myself.] The ratio of these two medians is only around 53.5% (1.07/2.0).

However, the main conclusion in the last sentence of the abstract and the last paragraph of the model results (Page 8), "the positive effect of this freshening is NEARLY STRONG ENOUGH TO CANCEL the suppressive effects of global warming-induced ocean thermal structure changes examined in previous studies", is not changed in the revision. The conclusion about the positive effect of the freshening is exaggerated, when the ratio of the positive effect to the negative effect is only around 50%-60%. The positive effect of the freshening is only around HALF of the suppressive effect of ocean thermal structure changes.

I suggest that: 1) the conclusion in the abstract and the model results must be revised; 2) the discussion section which states the two effects in parallel and implies the two effects are equal should explicitly state the relative importance of these two effects; and 3) the large intermodel spread and the apparent outlier model of the positive effect of the freshening should be appropriately discussed in the main text.

In addition, in the revised supplementary, the authors use more realistic PWP models to estimate the changes in ocean cooling effect induced by the freshening. It can be found that the ocean cooling effect calculated by Tdy-L is overestimated about 20% relative to the results by PWP. The different result is diluted in the supplement and is not mentioned in the main text. More importantly, the comparison between the positive effect of the freshening and the negative effect of the ocean thermal structure changes is not checked by PWP. This is the most important conclusion as the authors state it in the last sentence of the abstract. I suggest that the authors should estimate the difference of these two effects based on PWP and compare them with the results based on Tdy.

Reviewer #2 (Remarks to the Author):

I appreciate authors' replies to my comments and their revised version of the manuscript. The authors' replies sufficiently solved the issues and my concerns in the previous version of the manuscript. The manuscript is sufficiently revised and I have no further comments on this paper. I believe that the manuscript of the present version may be possible to be published.

Reviewer #3 (Remarks to the Author):

Overall, I feel that the points raised in the previous round of review have been satisfactorily addressed, but two points need to be slightly improved:

1- The first point raised in the previous round of review has been satisfactorily addressed, to the exception of this minor concern: In the last paragraph of the supplementary information, the authors write "unlike salinity, the mean change in DPI due to temperature is not statistically significant, with or without the contribution from MPI-ESM-LR". I disagree. Looking at SI Fig.6, we can see that the DPI anomaly induced by a change in temperature is negative for all the models, so it is statistically different from zero. If a t-test is conducted, it gives a rejection of the null hypothesis with a high significance, with or without MPI-ESM-LR.

2- I still need additional information for the second response. First, it is maybe not essential, but why only considering the Northern Hemisphere in Fig.R6? Then, why is the SSS signal not significant in the North Atlantic (still in Fig.R6)? Is there a model disagreement on the sign of the SSS response? All these important results should appear in the manuscript, or at least in SI.

We would like to thank the editor and all three reviewers for their constructive and positive comments that helped improve our manuscript. Here's our point-by-point response to your comments.

Reviewer #1:

The most attractive conclusion in the original draft is that “the positive effect of this freshening is NEARLY STRONG ENOUGH TO CANCEL the suppressive effects of global warming-induced ocean thermal structure changes examined in previous studies.” This conclusion is roughly acceptable, when the positive effect of the freshening is around 88.3% (1.43/1.62) of the suppressive effect of ocean thermal structure changes in the original draft. This conclusion was made based on only nine members in the original submission.

In the revision, the authors use a larger group of ensemble members, 18, to estimate the effect of upper-ocean freshening. However, the ratio of these two effects is only around 63.8% (1.18/1.85) in the revised draft, when more models are considered. The large decrease of the positive effect is due to the fact that there is an apparent outlier model in the original nine models, MIROC-ESM, with extremely high effect of SSS changes around 4.7. When more models are included, the extraordinary contribution of MIROC-ESM is reduced.

If we exclude this apparent outlier model, the mean of the other 17 models is only around 0.97, decreasing around 20%. The authors avoid this important issue in the main text. In the supplementary materials, the authors discuss this issue but state that there is also a model with a relatively large effect of ocean thermal structure changes. Of course, there must be one model possessing the largest value. If we exclude the model with the largest suppressive effect, the mean of the other models are around 1.67. Accordingly, the ratio of these two effects further decreases to 58% (0.97/1.67).

Another metrics for a group of values is the median of the values. The model with the ninth largest positive effect of the freshening among the 18 models is HadGEM-CC with a positive effect around 1.07, whereas the model with the ninth largest suppressive effect of is CMCC-CM with a negative effect around 2.0 [The values are roughly measured from Fig. S6 by myself.] The ratio of these two medians is only around 53.5% (1.07/2.0).

However, the main conclusion in the last sentence of the abstract and the last paragraph of the model results (Page 8), “the positive effect of this freshening is NEARLY STRONG ENOUGH TO CANCEL the suppressive effects of global warming-induced ocean thermal structure changes examined in previous studies”, is not changed in the revision. The conclusion about the positive effect of the freshening is exaggerated, when the ratio of the positive effect to the negative effect is only around 50%-60%. The positive effect of the freshening is only around HALF of the suppressive effect of ocean thermal structure changes.

I suggest that: 1) the conclusion in the abstract and the model results must be revised; 2) the discussion section which states the two effects in parallel and implies the two effects are equal should explicitly state the relatively importance of these two effects; and 3) the large intermodel spread and the apparent outlier model of

the positive effect of the freshening should be appropriately discussed in the main text.

Thank you for bringing this up. In our manuscript, we reach our conclusions based on both observations as well as model results. In observations, the change in DPI due to salinity (~ 0.2 m/s) overwhelms the change in DPI due to temperature (~ -0.13 m/s) by a comfortable margin after accounting for natural variability (ENSO/PDO). On the other hand in the CMIP5 analysis, in agreement with the reviewer's observation, salinity negates about 50-60% of the change in DPI due to temperature. As a balance between the two, we suggested that changes in DPI due to salinity nearly cancel the changes in DPI due to temperature. However, considering the reviewer's suggestions, we modify our manuscript to explicitly mention the effect of salinity changes on DPI from both observations as well as from models. We also expand our discussion of intermodel spread and outliers in the main text. (main text and figure 4 of the main manuscript)

In addition, in the revised supplementary, the authors use more realistic PWP models to estimate the changes in ocean cooling effect induced by the freshening. It can be found that the ocean cooling effect calculated by Tdy-L is overestimated about 20% relative to the results by PWP. The different result is diluted in the supplement and is not mentioned in the main text. More importantly, the comparison between the positive effect of the freshening and the negative effect of the ocean thermal structure changes is not checked by PWP. This is the most important conclusion as the authors state it in the last sentence of the abstract. I suggest that the authors should estimate the difference of these two effects based on PWP and compare them with the results based on Tdy.

Thanks for noting this. Based on the PWP simulations conducted previously, we showed that, on average, the mixing length from our formula is larger by about 20% compared to the PWP mixed layer depth. However, as we mentioned previously, the mixing length in our formula is the depth to which mixing is confined, while there is active mixing beneath the PWP mixed layer. For this reason the PWP mixed layer cannot be objectively compared with our mixing length. The parameter that ultimately matters for DPI computation is the cold wake and our previous analysis shows that cold wakes estimated from our formula are in good agreement with those from PWP. Also, using direct observations of satellite microwave SSTs, we show that our formula's cold wakes are in excellent agreement with the observed cold wakes.

However, to test the robustness of our conclusions, we performed a sensitivity analysis using output from the CMCC-CM climate model, which approximately represents the CMIP5 multi-model mean in terms of the DPI changes. Previously, we showed that cold wakes from PWP are stronger than our formula's cold wakes by $\sim 14\%$ when salinity is included, and by 5% when salinity is not included. To evaluate the impacts of such differences between PWP and our formula on estimating salinity effects, we scaled our formula's cold wakes to match the PWP cold wakes, with and without salinity, and re-computed the DPI changes. Results reveal that the contribution of salinity changes to DPI changes is 53% of the contribution from temperature changes, after making this

adjustment. Previously, the contribution of salinity changes to DPI changes estimated from CMCC-CM based on our formula was about 51% of the contribution from temperature changes. Thus, our main conclusions based on our formula's cold wakes are fairly robust.

Reviewer #3

Overall, I feel that the points raised in the previous round of review have been satisfactorily addressed, but two points need to be slightly improved:

1- The first point raised in the previous round of review has been satisfactorily addressed, to the exception of this minor concern: In the last paragraph of the supplementary information, the authors write "unlike salinity, the mean change in DPI due to temperature is not statistically significant, with or without the contribution from MPI-ESM-LR". I disagree. Looking at SI Fig.6, we can see that the DPI anomaly induced by a change in temperature is negative for all the models, so it is statistically different from zero. If a t-test is conducted, it gives a rejection of the null hypothesis with a high significance, with or without MPI-ESM-LR.

Thanks for noting this. Although the changes in DPI due to temperature from various models have the same sign, the mean DPI change is not statistically significant at the 95% level (t-value of 0.94) due to the larger standard deviation. We have checked our calculation of the t-statistics to confirm this.

2- I still need additional information for the second response. First, it is maybe not essential, but why only considering the Northern Hemisphere in Fig.R6? Then, why is the SSS signal not significant in the North Atlantic (still in Fig.R6)? Is there a model disagreement on the sign of the SSS response? All these important results should appear in the manuscript, or at least in SI.

Thanks for bringing this up. We noticed an error in our plotting program for the previous figure R6. Sorry for the confusion. We rectified this error and re-analyzed results from the entire Northern Hemisphere. Our analysis reveals that there are statistically significant changes in surface salinity for both the Central-Eastern Pacific as well as the North Atlantic. Averaged over the region 180W – 120W and 5N-25N, the surface salinity reduces by -0.038 psu/decade, a value significant at the 95% level. Over this region, the freshening of the surface ocean increases the DPI by 0.115 m/s/decade. On the other hand in the North Atlantic, averaged over the region 70W-40W and 10N-25N, the salinity increases by 0.055 psu/decade, a value significant at the 90% level. However, unlike the Pacific, the changes in DPI due to salinity in the Atlantic are not statistically significant. We now add a figure and discuss these results in the main manuscript (figure 5 and discussion section of the main manuscript).

REVIEWERS' COMMENTS:

Reviewer #1 (Remarks to the Author):

My previous concerns have been well addressed. I have no further comments on this manuscript. I think the present manuscript is suitable to be published.

Reviewer #3 (Remarks to the Author):

The authors have addressed my comments, and I do not have additional remarks. From my point of view, the draft is suitable for publication in Nature Communications.